

# PetPals
## A social app for dog owners

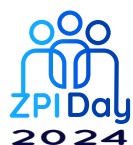

**Autors**: Aleksandra Broda◉ · Dominika Kołowrotkiewicz◉ · Dominika Sachanbińska◉
**Supervisor:** Michał Szczepanik

### Abstract

PetPals is a mobile application designed to address the problem of urban isolation among dog owners. The primary objective of the project was to develop an accessible, user-friendly platform that enables users to organize group walks, connect with others, and track their activity. The created app features real-time geolocation to find nearby users on a walk, customizable dog profiles to engage users, and physical activity tracking to promote healthier lifestyles for pets and their owners. The user-centred design of PetPals prioritizes simplicity and accessibility, ensuring smooth on-the-go use during walks. By bridging the gap between community building and pet care technology, PetPals aims to create a positive impact, encouraging healthier habits, stronger social bonds, and a more connected experience for dog owners everywhere.

## 1   DEVELOPMENT

### 1.1   Introduction

Urban isolation, a phenomenon characterized by feelings of loneliness and disconnection in densely populated environments, is a growing issue in modern society. Studies such as Long et al. (2024) reveal how loneliness and limited social support networks are more pronounced in urban settings compared to rural areas [9]. Other findings by Burger et al. (2020) suggest that urban residents often report lower happiness levels compared to their rural counterparts, primarily due to weaker social bonds and a lack of community integration [13].

As members of the dog owner community, we have observed that urban isolation extends to dog owners as well. While owning a dog can provide companionship and opportunities for social interaction, the fast-paced nature of urban environments often prevents meaningful connections among owners [12]. This lack of connection limits the potential mental health benefits of pet ownership.

Another significant challenge PetPals aims to tackle is the lack of convenient and widely-known tools for dog owners to find and join group walks in an organized, widely-available manner. While some rely on dedicated Facebook groups to coordinate meetups, these platforms often lack a structured or personalized approach. Furthermore, many individuals do not use Facebook, leaving them excluded from such opportunities. In certain neighborhoods or parks, it's common for owners to bring their dogs at specific times to socialize and play. However, many people miss out on these informal gatherings, while for the others most of these interactions remain occasional or surface-level due to the absence of an easy, non-intrusive way to reconnect with the same owners in the future.

To mitigate these issues, our project aims to create a mobile-first platform specifically tailored for dog owners. Our primary objective is to address urban isolation by helping to develop a sense of community through shared experiences such as group walks and social features. Then, by combining functionalities like messaging and user profiles with a geolocation-based outdoor activity tracker, we seek to create a useful tool that not only improves social connections but also enhances the overall experience of dog ownership in urban settings.

From a technical perspective, PetPals focuses on delivering an accessible, user-friendly solution. The app leverages a mobile-first design, recognizing that dog owners are often on the go. It also integrates technologies like real-time map view and walk tracking to provide tangible health benefits and build stronger, happier neighbourhoods.

### 1.2   Related Work

#### 1.2.1   Market Analysis

The pet tech market is expected to grow at a compound annual growth rate of 5.1% [17]. Currently, it includes several mobile applications for dog owners, but few combine social features with activity

tracking like PetPals. Two notable competitors are BarkHappy and Meet My Doggo, each with distinct approaches that differ from ours.

BarkHappy, designed primarily for users located in the United States, focuses on finding dog-friendly locations on a map and encouraging users to meet there. Its social features include geolocation to organize gatherings and lost-and-found alerts. However, BarkHappy lacks advanced customization, such as detailed dog profiles, and its development appears inactive, with the app being unavailable on the Google Play Store (state as of 28 November 2024) [4].

Meet My Doggo adopts a different strategy by connecting dog owners with dog lovers who may not own pets but want to spend time with them. The app emphasizes building connections between people and dogs, but does not support group walks, activity tracking, or geolocation for real-time meetings [14]. Its niche appeal makes it unsuitable for broader community building.

In contrast, PetPals combines both the world of social networks and health care. This integrated approach addresses urban isolation while providing practical tools for dog care. Unlike our competitors, PetPals is designed for long-term connections and encouraging spontaneous social interactions.

### 1.2.2 Technology Choices

**REST API** For backend, we chose the REST API pattern due to its simplicity, scalability, and compatibility with a wide range of client platforms, including mobile apps, enabling efficient data transfer via stateless operations over HTTP [16].

**Spring Boot** To implement it, we selected Spring Boot as the main framework due to its consistent performance in handling high request loads as shown in a study by D. Choma et al. (2023). As illustrated in Figure 1, it demonstrates low response processing times across varying request levels. Additionally, Figure 2 highlights its low error rates, ensuring reliable request handling even under increased load. [7]

Furthermore, according to Stack Overflow (2024), it is one of the most popular backend framework on Stack Overflow with 12.7% of votes, providing access to extensive community support and resources, making it a reliable choice for backend development. [15]

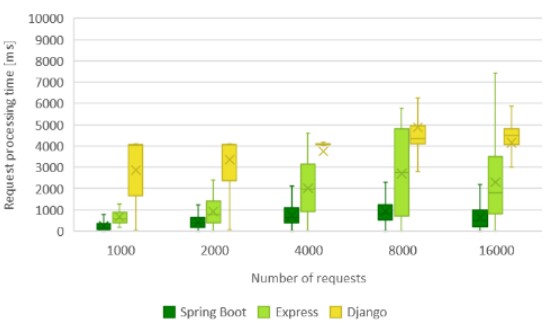

Figure 1: The GET request processing time depending on the number of requests for each of the tested frameworks.

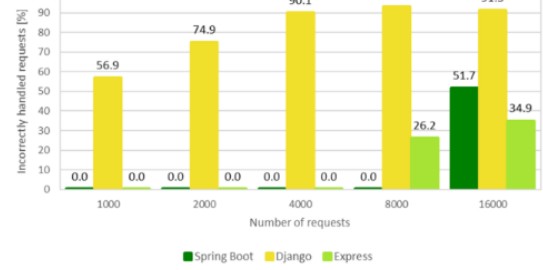

Figure 2: The percentage of incorrectly handled GET requests by the application based on a given framework.

**PostgreSQL** We chose to use a relational database for our project because it efficiently handles structured data with relationships between entities, such as users, dog walking sessions, and locations, making it ideal for managing and querying complex, interconnected data while ensuring data integrity and scalability [6].

It was also selected due to its recognition as the most popular database in the Stack Overflow survey, reflected in the strong community support [15].

Furthermore, according to research by A. Akhtar (2021), it demonstrates a steady linear growth pattern in popularity over time, indicating its reliability, adaptability, and increasing preference among developers and organizations. This combination of popularity and consistent growth makes PostgreSQL a robust and forward-looking choice for database management [1].

**AWS S3 Storage** For file storage, we chose AWS S3 due to its reliability, scalability, and compatibility with our application's requirements [18]. We utilize S3 in conjunction with presigned URLs, enabling secure, time-limited access to images. This approach minimizes server load by offloading file transfer operations to S3, ensuring efficient and secure handling of user-generated content. The use of presigned URLs allows us to maintain strict control over file access without requiring complex additional infrastructure, making S3 a practical and streamlined solution for our storage needs.

**React Native**   React Native framework was selected due to its combination of performance, development efficiency, and compatibility with our team's skill set. It provides near-native performance through the use of the Hermes engine, which pre-compiles JavaScript into byte-code, enabling smooth execution and addressing potential performance bottlenecks. While it may not match the UI rendering capabilities of native frameworks like Kotlin or the complex UI handling of Flutter, React Native delivers reliable performance for the majority of use cases, making it a practical choice for our application [2].

Furthermore, the team's familiarity with React and JavaScript played a crucial role in the decision. Previous experience with React Native development ensured a minimal learning curve, enabling the team to start development efficiently without the need for extensive retraining.

Finally, React Native's large and active community provides strong support and resources, and its use in widely adopted applications like Facebook, Instagram, Pinterest, and Microsoft Teams demonstrates its stability and scalability for a wide range of projects. These factors combined made React Native a well-suited choice for our app's development.

**Typescript**   When selecting the in-framework language, we chose TypeScript over JavaScript due to its ability to improve code quality and reduce runtime bugs as showcased in studies (Bogner, 2022). According to research, static typing in TypeScript helps catch up to 15% of bugs during development, compared to plain JavaScript, saving valuable time on debugging and ensuring faster delivery. Additionally, its improved maintainability and predictability make it ideal for scaling projects, aligning with our goals for efficient and reliable development [5].

**Expo**   For workflow management in React Native, we chose Expo due to its broad array of features tailored for rapid development. As the leading choice for React Native projects, with over 50% of all projects that use it, Expo is particularly suited to create MVPs such as our project due to its streamlined setup and efficient development workflow [8].

One of Expo's main advantages is its opt-in modules, which include solutions for navigation, background and foreground geolocation, image optimization, animations, and more, all of which significantly sped up development time. Additionally, tools like the Expo Go app make testing on target devices fast and straightforward, while Expo Application Services simplify deployment by managing the application build and release processes.

### 1.2.3   Limitations

**Time Constraints**   With a tight timeline beginning in early October and culminating in the final presentation on December 13th, the focus remained on delivering core functionalities included in the MVP while ensuring a user-friendly experience.

**Team size**   The project faced limitations due to having only three team members, impacting workload distribution and collaboration compared to a traditional four-member team.

**Budget**   Due to a limited budget, our team had to rely on free tiers or student versions of software and API access during development, which impacted the current scalability and feature availability of the developed application.

**Licensing**   Academic nature of the project resulted in some licensing constraints, which restricted team's access to certain proprietary tools and datasets, compelling us to either seek open-source alternatives or develop custom solutions.

**Access to Apple devices during development**   Due to the lack of access to iOS devices, our project was constrained to prioritize development and optimization for Android platforms, ensuring we maximized the available resources and expertise.

### 1.2.4   Resources

**AWS Student account**   As part of the development, the AWS Student Account provided access to essential cloud services like AWS S3 for secure file storage, enabling scalable infrastructure and reducing operational costs.

**Google Maps Platform Free tier**   Google Maps Platform's free tier was crucial for the location-based features of the app, such as geocoding, reverse geocoding, and real-time location tracking. It allowed us to integrate powerful mapping and navigation services without incurring significant costs during the development phase [10].

**Hive MQTT Cloud**   Hive MQTT Cloud was utilized to enable real-time communication for live location sharing in the app. As a lightweight and scalable messaging protocol, MQTT supported reliable data transmission, allowing users to share their walking routes and locations efficiently with minimal latency [11].

## 1.3   Results

### 1.3.1   Implemented Functionalities

**Social features**   Users can message other users and manage friend requests through a simple and intuitive interface.

**Authentication system**   Standard email and password authorization paired with JWT token-based security ensures secure and scalable user authentication for our app.

**User and pet profiles**   Users can create personalized profiles for themselves and their pets, including details such as breed, temperament, and preferences, allowing for walk interactions based on compatibility.

**Group walk management**   Users can create, schedule, and join group walks, fostering community building and making the app a social platform for dog owners.

**Real-time geolocation**   The app includes live location tracking, allowing users to find nearby dog owners and join group walks.

**Activity tracking**   PetPals tracks walks in terms of distance, time, and speed, enabling users to monitor their activity levels and promote healthier habits for themselves and their pets.

**Privacy controls**   Customizable settings allow users to choose who can see their location or interact with them, ensuring safety and comfort.

### 1.3.2   Business and technical objectives met

**Accessibility**   Developed an accessible, mobile-first platform with compliance to accessibility standards, including WCAG guidelines for contrast and button size, improving usability for users with visual or motor impairments.

**Intuitive navigation**   Achieved a seamless navigation structure based on Jakob Nielsen's usability heuristics, reducing cognitive load and improving ease of use.

**Filling market gap**   Positioned PetPals strategically in the market by filling the gaps left by competitors, such as focusing equally on social networking and activity tracking.

### 1.3.3   Practical applications and benefits

PetPals is designed to make life easier and more enjoyable for dog owners by addressing key challenges they face every day. For example, urban isolation and social anxiety often make it difficult for people to start conversations or new friendships, even if they see numerous people walking their dogs in the same neighbourhood.

PetPals removes the pressing need to have an excuse in order to start a meaningful conversation by offering a map of dog owners who want to connect and, therefore, share their real-time location during walk. This invites others to seek them out and spend more time together than they probably normally would during a random meeting. If this is what the user prefers, PetPals can help make walks a social and uplifting experience rather than a solitary one.

The app also gives users more confidence and peace of mind during their walks. Features like customizable privacy controls ensure that location sharing feels secure and optional, while group walks

provide the opportunity to explore new areas with others, knowing you're not alone. For people who might feel hesitant or disconnected in a big city, this creates a safer, friendlier experience.

For dogs, PetPals offers a unique way to improve both their physical health and their behaviour. By enabling users to organize or join walks with other dogs that share similar traits, such as size, temperament, or energy level, the app ensures playtime is more efficient, regular, and simply fun for the pets. This helps dogs burn off energy, practice socialization, and reduce behavioural problems that are often caused by lack of interaction or exercise.

Overall, PetPals transforms daily dog walks from a routine task into a relaxing and rewarding experience for both owners and their pets. It fosters a stronger sense of connection, combats urban isolation, and promotes happier, healthier lifestyles for everyone involved.

# 2 CONCLUSION

## 2.1 Conclusions

In our opinion PetPals marks a significant step toward building a practical and engaging app for dog owners, addressing key challenges such as urban isolation and the need for simple and usable social tools.

PetPals distinguishes itself among its competitors through features like group walk scheduling, real-time maps, posts with comments and customizable dog profiles.

One of PetPals' key advantages is its mobile-first design, crafted for use in real-world scenarios such as dog walks in challenging conditions like bright sunlight or while wearing gloves. The app also prioritizes accessibility, with its carefully considered UI design that incorporates design elements that allow users with vision or motor impairment to use it. We believe that this practical approach will result in greater user retention and satisfaction.

The most significant success of PetPals is its ability to create a sense of community. By bringing people together, the app is uniquely positioned to help build positive relationships between users, improving both their social lives, physical and mental health, and also the well-being of their dogs. This makes PetPals not just a tool but a go-to solution for connection and support in the lives of urban dog owners.

## 2.2 Future Directions

We see several opportunities for enhancement to further improve PetPals' functionality and user engagement:

**Gamification** To boost user retention and long-term engagement, gamification elements such as badges, milestone rewards, and leaderboards could be introduced. For example, users could earn badges for completing a certain number of walks or organizing group events. Research shows that gamified features increase retention rates by 30-50% [3], making this a rational next step for PetPals in order to achieve higher user retention rates.

**Expanding Community Features** Introducing more tools for local event organization could enrich the sense of vast and diverse community within the app. We could think of ways to help professionals like dog walkers or trainers incorporate their services into the app. We could also help users host or participate in events other than dog walks, like training sessions, or longer, multi-day trips to make PetPals the go-to app for all dog-targeted functionalities.

**Integration with Third-Party Services** PetPals shows great potential to become an intuitive, multipurpose tool for dog owners. Partnerships with services such as veterinarian clinics, dog trainers, and pet supply stores could be very beneficial, both for the app and for the users. Features such as in-app reminders for vet appointments or discounts on pet products could provide additional value to users while opening potential revenue streams.

**Support for iOS** Expanding the app's reach by making it compatible with iOS devices could greatly increase the user base and accessibility.

**Add Support for Other Pets** The app could also include support for other pets, such as cats, rabbits, or birds, allowing users to connect with a broader community of animal owners.

**User Tips and Helpful Articles**   Including educational articles on pet care, walking tips, and health could add value to the platform for pet owners.

**Map of Dog Friendly Places**   Introduce a feature for marking dog-friendly or dangerous areas, helping users navigate better and share local insights.

**Diverse Post Formats**   Support various post types (e.g., images, polls, video) to make user interaction more engaging.

**Family Accounts**   Allow multiple users within a family to manage and share a single account, promoting family-oriented pet care.

**Scalability**   Improve app scalability by transitioning from WebSockets to a more robust solution like Kafka or a cloud-based messaging system to handle multiple database servers, enabling smoother operation under heavy traffic.

**Add Pictures to Walks**   Allow users to add photos to their walks, making profiles more engaging and visually appealing.

**Integrate OAuth**   Simplify the login process by integrating OAuth for social media logins, improving user experience and security.

## 2.3   Acknowledgements

The authors sincerely express their gratitude to Michał Szczepanik, Ph.D., our project supervisor, for his invaluable guidance, encouragement, and support throughout the project. The authors also extend their heartfelt thanks to Wojciech Thomas, Ph.D., and Oleksandr Yeroshkin, Ph.D., for their insightful feedback and assistance, which significantly contributed to the progress and success of this work.

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
