# OpenReview forum: "PetPals - a social app for dog owners"
_pwr.edu.pl/Wrocław_University_of_Science_and_Technology/2024/ZPI_Day — Wrocław University of Science and Technology 2024 ZPI Day Submission_

### Official Review · Reviewer_FZ5w · 2024-12-05
**PetPals - a social app for dog owners**

**Confidence:** 5
**Significance Of Results:** 5
**Overall Quality:** 5

**Compliance With Template:**

5: Very High Quality – The article contains all the required sections, which are written in a very detailed, clear, and error-free manner. The structure is professional and meets expectations, and the content adheres to the highest substantive and formal standards.

**Description Of Results:**

5: Very High Quality – The results are described in detail, clearly and comprehensively, supported by thorough evaluation, analysis, and convincing usage examples. The description meets the highest substantive standards.

**Feedback On Consistency:**

The project description is clear and well-structured. It begins with a strong problem analysis that effectively sets the stage for the technical implementation and results. The alignment between the identified problem—urban isolation among dog owners—and the proposed solution through PetPals is logical and convincing. The presentation of results, particularly the app’s functionalities and business objectives, is consistent with the initial goals of the project. The conclusion ties everything together effectively, reflecting on the achievements. However, improving the transitions between the technical implementation and user-centric elements and additional app screenshots would make the narrative more seamless and engaging.

The combination of a user-centered approach, strong technical foundation, and high-quality delivery by a small team makes PetPals a standout project with substantial potential for both social and business impact.

**Potential For Development:**

The app has significant business potential in a growing market. Features like gamification, community tools, and third-party integrations (e.g., with veterinarians or pet supply stores) could increase user engagement and generate additional revenue streams. The inclusion of educational content and expanded pet species support would also broaden its appeal. As a pet owner, I see real-world value in these developments and believe they could position the app as a leading solution in its niche. Furthermore, the high level of maturity demonstrated in the current implementation suggests the team is well-equipped to tackle these enhancements in future updates. Providing more specific examples of how these features could be implemented would make the development plan even stronger.

**Project Nature Evaluation:**

As a pet owner myself, I can appreciate the app’s innovative approach to addressing common challenges faced by dog owners. The project exhibits key engineering qualities, including the application of robust and scalable technologies like React Native, PostgreSQL, and AWS S3. These choices demonstrate thoughtful planning and technical maturity. Notably, the app balances technological complexity with user-focused design, ensuring practical utility for a diverse audience. Given that this project was developed by only three students instead of the standard four, the maturity and comprehensiveness of the solution are particularly impressive. This highlights not only the technical skills of the team but also their ability to manage resources effectively.

**Technical Language Precision:**

5: Very High Quality – The language is entirely appropriate for a technical report. All terms are used correctly and precisely, and the style is professional, clear, and coherent, without any errors or ambiguities.

---

### Official Review · Reviewer_KHMN · 2024-12-06
**PetPals - a social app for dog owners**

**Confidence:** 3
**Significance Of Results:** 4
**Overall Quality:** 4

**Compliance With Template:**

5: Very High Quality – The article contains all the required sections, which are written in a very detailed, clear, and error-free manner. The structure is professional and meets expectations, and the content adheres to the highest substantive and formal standards.

**Description Of Results:**

2: Low Quality – The results are described very superficially and in a general manner. Essential details, usage examples, or evaluations are missing.

**Feedback On Consistency:**

The information presented is logically related and creates a coherent text. However, the results are not supported by evidence that confirms their reliability and practical usefulness. The implemented functionalities are only briefly noted.

**Potential For Development:**

The project can be a basis for further development. Directions and ideas for further work have been described in detail.

**Project Nature Evaluation:**

The project results have practical applications, can be used in real conditions and contribute to solving specific problems.

**Technical Language Precision:**

5: Very High Quality – The language is entirely appropriate for a technical report. All terms are used correctly and precisely, and the style is professional, clear, and coherent, without any errors or ambiguities.

---

### Official Review · Reviewer_biDd · 2024-12-09
**PetPals - a social app for dog owners**

**Confidence:** 3
**Significance Of Results:** 4
**Overall Quality:** 4

**Compliance With Template:**

4: High Quality – The article contains all the required sections, which are well-written and substantively correct, although minor errors or shortcomings may be present. The overall structure is clear and coherent.

**Description Of Results:**

4: High Quality – The results are described in detail and supported by usage examples or evaluations. The description is reliable but may lack full depth of analysis.

**Feedback On Consistency:**

Opis projektu jest spójny, a prezentowane informacje są w większości logicznie powiązane, niemniej brak jest jednoznacznego wyodrębnienia/podkreślenia celu, czego czytelnik musi się doszukiwać w treści.

**Potential For Development:**

Artykuł wskazuje jakie aspekty projektu wymagają rozwoju w przyszłości. Zaproponowany system ma potencjał wdrożenia komercyjnego.

**Project Nature Evaluation:**

Projekt ma nie do końca wyraźny cel inżynierski, został określony sposób osiągnięcia celu co pozwala na jego zidentyfikowanie. Zastosowane metody oraz rozwiązania są odpowiednie do realizacji tego celu.

**Technical Language Precision:**

4: High Quality – The language is appropriate for a technical report. Terminology is used correctly, and statements are precise, with only minor shortcomings that do not affect the overall clarity.

---

### Decision · Program_Chairs · 2024-12-10

Accept (Poster)